# Providing personal information to the benefit of others

Bettina Rockenbach[1,2]*, Abdolkarim Sadrieh[3], Anne Schielke[1]

1 Department of Economics, Experimental and Behavioral Economics, University of Cologne, Cologne, Germany, 2 Department Experimental Economics, Max Planck Institute for Research on Collective Goods, Bonn, Germany, 3 Faculty of Economics and Management, University of Magdeburg, Magdeburg, Germany

☯ These authors contributed equally to this work.
* bettina.rockenbach@uni-koeln.de

## Abstract

Personal information is a precious resource, not only for commercial interests but also for the public benefit. Reporting personal location data, for example, may aid efficient traffic flows and sharing one's health status may be a crucial instrument of disease management. We experimentally study individuals' willingness to contribute personal information to information-based public goods. Our data provide evidence that—compared to monetary contributions to public goods—information may be substantially under-provided. We show that the degree of information provision is strongly correlated to the information's implicit (emotional and cognitive) costs. Individual's reluctance to share personal information with high implicit, in particular emotional costs, may seriously limit the effectiveness of information-based public goods.

## Introduction

The creation of public benefits frequently depends on individuals' willingness to provide personal information. The effectiveness of public disease control, for example, crucially depends on individuals' willingness to report a suspected illness to a health authority. Likewise, the accuracy of policy measures critically depends on individuals' willingness to report their socio-demographic data, and the exactness of traffic information importantly depends on individuals' willingness to share their current location. Advances in data collection and data mining enable far-reaching analyses and tremendously improved possibilities for inference. However, their success in creating a public good on the basis of the collected individual information crucially relies on individuals' willingness to provide personal information.

The last decades have generated an extensive–mostly experimental–literature on the willingness to contribute to the creation of public goods when costs and benefits are monetary. In how far the gained insights can be transferred to situations in which the unit of provision is personal information is unknown yet. Doubts apply on the basis of privacy concerns that may result in a lack of unbiasedness and completeness of the collected data. It has been shown that people hold idiosyncratic preferences for privacy, and that these preferences strongly depend on the context and the type of personal information [1–2]. Although the topic's relevance amplifies with the rapid technological advancements, to the best of our knowledge no study

**Data Availability Statement:** Data available at: https://dx.doi.org/10.23663/x2643.

**Funding:** Bettina Rockenbach and Anne Schielke acknowledge financial support from the German Research Foundation (DFG) through the research unit "Design & Behavior" (FOR 1371). www.dfg.de

The funder had no role in study design, data collection and analysis, decision to publish, or preparation of the manuscript.

**Competing interests:** The authors have declared that no competing interests exist.

investigates the willingness to provide personal information to the public benefit (the existing experimental literature focuses on trade-offs between individual costs and benefits, e.g., [3–9]. However, developing effective policy measures requires understanding the mechanisms governing information provision (e.g., [10]).

In this paper, we experimentally study the willingness to provide personal information to information-based public goods, and compare this to the willingness to provide money to material public goods. We set up four treatments, varying the unit of provision (*info* vs. *money*), and the explicit net cost of provision (*cost* vs. *no cost*). In the *info* treatments, subjects provide real personal information about themselves (e.g., about their preferences, past behavior or physical characteristics). Beyond explicit transaction or opportunity costs, the provision of personal information may incur implicit provision costs. One source of implicit provision costs may be cognitive costs of information compilation. Another source may be emotional costs, resulting from expected negative effects of information leakage or fear of ostracism, self-image concerns or disutility from cognitive dissonance. These implicit costs may induce heterogeneity in the preferences for provision. To study whether and how the implicit costs of information provision influence the willingness to provide information-based public goods, we systematically vary these costs. In a pre-experimental survey, we ask participants to rate how cognitively difficult and how emotionally demanding they find answering questions on particular personal information. The responses of these participants (a separate set of subjects from the same subject population as in our experimental) provide us with a ranking of our questions by their implicit cost of information provision.

Our experimental results show that there is a structural difference between information and money provision to the benefit of others. Personal information that ranks low in implicit provision costs is provided much more frequently than information that ranks high. Moreover, emotional costs seem to loom larger than cognitive cost. As a result, when monetary cost of contribution is zero, selective information provision may lead to lower public good levels with information contributions compared to the levels achieved with monetary contributions.

Our study provides evidence that even in an abstract laboratory setting, where subjects' privacy and data protection can be fully secured and the material cost of contributions can be eliminated, subjects incur implicit costs of information provision. This underlines the importance of the implicit costs of information provision in information public goods. Our study demonstrates that structural distortions in the level of information public goods should be expected, depending on the implicit cost of information provision that the underlying information requests incur for the contributors.

## Related literature

In recent years, various disciplines have contributed to the growing literature on privacy. A survey by [2] illustrates the broad scope of the issue that touches several disciplines, e.g., the legal sciences, philosophy, computer sciences, and economics. In economics, part of the literature focuses on the *informational* aspects of privacy taking a regulatory perspective, while another part studies the *behavioral* aspects of privacy that arise due to the trade-off between privacy concerns and benefits of information provision. The latter literature is most closely related to our study.

[1–2] summarize the empirical evidence of the literature on the behavioral aspects as follows: First, the implicit cost of information provision is context-dependent for most individuals, i.e., the same person may provide personal information in one situation, while not doing so in another. Second, the implicit costs of information provision are idiosyncratic, i.e. in any given situation, some individuals will provide their personal information while others will not.

The extent of information provision varies with the type of information required, the framing of the situation, and individual characteristics of the decision maker. For example, [11] observe a lower likelihood of information provision when respondents are presented newspaper articles with negative compared to positive aspects of companies' privacy policies. [12] also uncover framing effects showing that the value that subjects assign to privacy protection depends on their initial endowments and the ordering of choices. [13] find that subjects' willingness to sell the information on sensitive issues (i.e., their body weight) is on average higher than their willingness to pay to protect that information.

A number of authors observe individual differences in the monetary value assigned to private information. In an early study, [9] find that subjects with socially less desirable traits demand higher prices for their personal information than others do. In another experiment, [5] show that a positive probability of non-anonymous feedback on IQ tests induces the subjects with below-median results to sell less personal information. [4] observe that the likelihood of information provision is higher (1) the less embarrassing the information is and (2) the more socially distant the recipients are. In an experimental study on selective information provision, [3] also observe considerable heterogeneity in subjects' preferences for privacy, depending on the type of information, and show that anonymity increases the likelihood of information provision.

Some studies show that the willingness to trade benefits (e.g. lower prices) against the provision of private information is relatively high in many purchase situations, but depends on the privacy policies of the retailers. For example, [8] find that the likelihood of a purchase and the willingness to pay are higher if the retailer engages in privacy protection. [7] observe that a non-negligible fraction of moviegoers is willing to purchase from the retailer with the higher price if that retailer requests less personal information or promises not to use the information for marketing purposes. In a different field experiment, [6] find almost no effect of privacy protection. Examining the willingness to purchase from two competing online retailers with different privacy policies, they find subjects generally prefer to purchase from the retailer with the lower price.

Our experimental study goes beyond the existing literature on the provision of personal information, because we examine how social benefits affect the information provision, while the existing literature has generally focused on the private benefits. However, our hypotheses are informed by the literature, in a number of ways. Given the existing results, we presume that the willingness to provide personal information will vary both across individuals and across the type of information requested. Furthermore, we presume that increasing the monetary cost of information provision will negatively affect the provision of all types of information, because the existing literature suggests that individuals generally engage in a cost-benefit assessment when deciding whether to provide information or not.

## Voluntary information provision to the benefit of others

The aim of our study is to investigate whether there is a fundamental difference between 'money' and 'information' as units of provision to a public good. To serve this more foundational character we abstract from any specific application. And, importantly, to have both conditions (*money* and *info*) as parallel as possible, in the *info* condition the public good is not created by the aggregation of the specific information provided (like in a puzzle), but by the *willingness* to provide information. Providing information (as providing money) creates a monetary benefit to others, which does not depend on whether or not the information is truthful, but only whether or not people are willing to provide. This may be seen as a proxy for real world applications where truthful reporting is not a critical point, because the info is gathered automatically, e.g., location data or search tracing.

## The game

*Public good with money provision* We set up a public goods game with $n \geq 2$ players (indexed $i = 1, \ldots, n$). Each player is endowed with $e$ monetary units. Players simultaneously decide how many of these units of endowment to provide to the public good. Player $i$'s provision is denoted by $g_i \in \{0, \ldots, e\}$. For each unit player $i$ provides, she incurs an explicit net monetary provision cost of $c \geq 0$. Each unit provided by the $n-1$ other players increases player $i$'s utility by $0 < a < 1$. Let $g = (g_1, \ldots, g_n)$ denote the vector of provision decisions of the $n$ players. Then, player $i$'s utility function in the game with money provision is given by:

$$U_i(g) = e - cg_i + a \sum_{j \neq i} g_j. \tag{1}$$

There is an important structural difference between the provision of money and the provision of information: While the units of provision are not distinguishable in the game with money provision, the units of provision may have different implicit provision costs in the game with information provision (see discussion in section 2). That is, in addition to the explicit net cost of provision, player $i$ may incur an implicit cost of provision if she provides personal information. This implicit cost may differ for different types of information. We account for this by adding an additional cost term $\gamma_{ik}$ to the player's utility function and allow the units of provision to be distinguishable.

*Public good with information provision* In the game with information provision, each player $i$ is endowed with a set of items of private information $\Theta_i = \{\theta_{i1}, \theta_{i2}, \ldots, \theta_{im}\}$. We assume that the number of items $m$ in a player's information set $\Theta_i$ is identical for all $n$ players. Each player $i$ receives utility $v_{ik}$ from item $\theta_{ik}$ (e.g., knowing the own preferences, past behavior or physical characteristics is valuable for the individual's current and future decision-making), that is her endowment is $e_i = \sum_{k=1}^{m} v_{ik}$. Let $x_{ik} \in \{0,1\}$ be $i$'s choice variable that indicates whether player $i$ provides the $k$-th item $\theta_{ik}$ of her information set $\Theta_i$, with

$$x_{ik} = \begin{cases} 0 & \text{if player } i \text{ does not provide } \theta_{ik} \\ 1 & \text{if player } i \text{ provides } \theta_{ik}. \end{cases}$$

Thus, player $i$'s provision decision is given by the choice vector $x_i = (x_{i1}, x_{i2}, \ldots, x_{im})$. Let $x = (x_1, \ldots, x_n)$ denote the vector of provision decisions of the $n$ players. Then, player $i$'s utility function in the game with information provision is given by:

$$U_i(x) = e_i - \sum_{k=1}^{m}(c + \gamma_{ik})x_{ik} + a \sum_{j \neq i} \sum_{k=1}^{m} x_{jk}. \tag{2}$$

The utility function, for most of its parts, corresponds to the standard objective function in public goods settings. The structural difference is that we incorporate an additional cost variable $\gamma_{ik}$ to capture the implicit cost of information provision, such as e.g., cognitive and emotional cost, that may vary between players $i$ as well as between items $k$. In general, we expect the implicit provision cost to be linked to the cognitive and emotional load of the requested information and to rank across items according to the rankings that we elicit in our pre-experimental survey. Note that participants receive no feedback on the aggregated information and thus cannot gain additional benefits or costs from aggregated info. To make info and money provision comparable, our experiment only uses one outcome dimension, which is money.

In the game with information provision, players simultaneously decide for each item $\theta_{ik}$ of their information set $\Theta_i$ whether to provide the item or not. We assume that the provision of an item does not reveal the personal information to any other player. As in the game with

money provision, player $i$ incurs an explicit (monetary) net provision cost of $c \geq 0$ that is identical for all items. Each item provided by the $n-1$ other players increases player $i$'s utility by $0 < a < 1$.

All differences between Eqs (1) and (2) stem from two structural differences between money and information: First, the units of provision are distinguishable in the case of information, while this does not hold for money. This is captured by the indicator choice variable $x_{ik}$ and the individual value of information $v_{ik}$ in Eq (2). Second, we include an additional cost of provision, the implicit cost $\gamma_{ik}$, in Eq (2). All other model parameters, especially the net cost from the own provision $c$ and the return from another player's provision $a$, do not differ between the two models.

## Hypotheses

A rational player $i$ maximizes her utility with respect to the items she provides. Obviously, as long as there is a positive net cost to the provision of information (no matter whether explicit or implicit) the dominant strategy is not to provide any information. Hence, in case of positive costs, there is a unique equilibrium in which neither player provides any information. If both the explicit net costs and the implicit provision costs are zero ($c = 0$ and $\gamma_{ik} = 0$), players are indifferent between providing and not providing information. Then any mixture of provision and non-provision may be in equilibrium. The equilibria in the game with monetary contributions have similar characteristics. In the equilibrium with explicit net costs of provision, players do not provide any money to the public good, while with zero explicit net costs, any level of provision is possible. If we assume that players are concerned about others' material payoffs, it is likely that Pareto-efficient equilibria are selected more frequently.

## Hypothesis 1

> *Both in the game with info and the game with money provision, the provision level with positive explicit net provision costs is lower than with zero explicit net provision costs.*

Our second hypothesis concerns the difference between information and money provision. Conceivably the provision of information involves higher costs than the provision of money, since any implicit costs of information provision top off the explicit net costs. Hence, ceteris paribus, we expect higher levels of money than information provision, assuming that the implicit costs are not zero for all items.

## Hypothesis 2

> *The provision level in the game with money provision is higher than in the game with information provision.*

In the case in which players hold social preferences the predictions of hypothesis 1 remain unchanged. This is true because the explicit net costs of provision generally drive the cost-benefit calculus of players towards less provision, no matter whether or not they hold additional other-regarding preferences. However, if a player maximizes the sum of all group players' utilities, even items with a positive implicit provision cost may be provided. From the utility function in Eq (2) we see that in the social optimum of the game, player $i$ provides all items with certainty that exhibit zero implicit provision costs (which is in contrast to the set of Nash equilibria under $c = 0$ where player $i$ is just indifferent with regard to provision). To maximize joint

**Table 1. Treatments.**

|  | *info* | *money* |
|---|---|---|
| *cost* | INFO | MONEY |
| *no cost* | INFO_NC | MONEY_NC |

utility, she also provides items with a low implicit provision cost ($\gamma_{ik} < -c + a(n-1)$). This means that if a player holds social preferences and incurs only low implicit costs of information provision, she is more likely to provide more information with low than with high implicit costs.

## Hypothesis 3

*The likelihood of provision is higher for information with low implicit costs than for information with high implicit costs.*

## The experiment

### Experimental design

We investigate the voluntary provision of personal information to the benefit of others, and compare it to the voluntary provision of money. To examine whether there are structural differences between the provision of money and information, we set up four treatments in a 2x2-design where we vary the unit of provision (*info* vs. *money*) and the explicit net monetary provision costs (*cost* vs. *no cost*). The four treatments are given in Table 1:

All treatments are one-shot paper-and-pencil experiments with a group size of four players. In the money treatments, subjects are endowed with 20 monetary units and decide how many units to contribute to the public good. In the info treatments subjects receive 20 questions and decide which of them to answer. Answering a question ("providing an item") means providing to the information-based public good. Each item and each unit is worth €0.30, i.e., the total endowment of each subject equals €6.00 = 20×€0.30. The decision to keep an item (i.e., not provide the information) or a monetary unit gives the player €0.30, while each of the other players receives zero, both in the *cost* and the *no cost* condition. The decision to provide an item (i.e., provide the information) or a unit gives €0.12 to the provider and each of the other players in the *cost* condition, and hence incurs a net monetary provision cost of €0.18. In the *no cost* condition, the decision to provide an item or a unit does not change the payoff of the provider, but gives €0.12 to each of the other players. Hence, the net monetary provision cost is zero in the *no cost* condition. We decided to implement this zero cost condition in addition to our cost condition to investigate whether or not there is a difference between money and information provision under this boundary condition of true monetary indifference. By comparing provision levels under the different net monetary costs of provision, we can investigate whether information provision can be incentivized in the same way as money provision. In Table 2, the marginal payoffs from the own provision and from the provision of the other

**Table 2. Marginal payoffs.**

|  | *cost* condition | | *no cost* condition | |
|---|---|---|---|---|
|  | **self** | **others** | **self** | **others** |
| item/unit kept | €0.30 | €0.00 | €0.30 | €0.00 |
| item/unit provided | €0.12 | €0.12 | €0.30 | €0.12 |

group members are given for the *cost* and the *no cost* condition (marginal payoffs are identical for the *info* and the *money* condition).

## Implicit provision costs

In the light of the more fundamental nature of our study, our questions are not meant to mirror a specific real-life application, but are selected to be informative for certain characteristics of gathered information. Nonetheless, some information is very close to what is gathered in real-life applications, like the questions on objective information such as age, gender, or zip-code or information on preferences such as favorite song, drink, movie or actor. Other information may be retrieved more indirectly, e.g., BMI through size and height in online shopping. As the provision of information may incur implicit provision costs (in addition to the explicit costs), we assess a proxy for the implicit costs of the 20 items (questions) in a survey study in a large undergraduate economics class where we focus on the dimensions of cognitive and emotional costs (see S1 File). A total of 211 respondents separately evaluated each of the 20 items, concerning the *cognitive load* ("When answering this question I have to think. . .") and the *emotional load* ("When answering this question I feel. . ."). Answers to the cognitive load are on a 6-point Likert scale from (". . .I have to think. . .") "very little" to "very hard". Answers to the emotional load are on a 6-point Likert scale (". . .I feel. . .") "very uncomfortable" to "very comfortable". We find no order effects in the evaluation of items. Spearman's rank correlation coefficient between the order of presentation and the mean and median item assessment is insignificant both for the cognitive and emotional load (p-values range from $p = 0.5908$ to $p = 0.8750$). Table 3 presents the mean and median item evaluation, where "very little" and

**Table 3. Items in the *info* condition.**

| No. | Item | Question | Cognitive load | | Emotional load | | Combined measure |
|---|---|---|---|---|---|---|---|
| | | | Mean | Median | Mean | median | |
| 1 | gender | Are you male or female? | 1.0948 | 1 | 1.4976 | 1 | 0.5066 |
| 2 | eye color | What is your eye color? | 1.4787 | 1 | 1.5024 | 1 | 0.6939 |
| 3 | age | What is your age? | 1.1848 | 1 | 1.7062 | 1 | 0.7300 |
| 4 | subject of study | Are you currently enrolled and if so, what is your subject of study? | 1.1517 | 1 | 1.8057 | 1 | 0.8198 |
| 5 | shoe size | What is your shoe size? | 1.4739 | 1 | 1.8246 | 1 | 0.9511 |
| 6 | study duration | If you are studying, how long have you been studying so far? | 1.3365 | 1 | 1.9336 | 2 | 0.9924 |
| 7 | height | What is your height? | 1.4455 | 1 | 1.9526 | 2 | 1.0516 |
| 8 | zip code | What is your zip code? | 1.6825 | 1 | 1.9621 | 1 | 1.1796 |
| 9 | clothes | How often do you return clothing to the seller as unused after having actually used it? | 1.8057 | 1 | 1.9573 | 2 | 1.2513 |
| 10 | credit | How often do you overdraw your bank account? | 1.5071 | 1 | 2.1659 | 1 | 1.2714 |
| 11 | season | What is your favorite season? | 2.1754 | 2 | 1.6635 | 1 | 1.3497 |
| 12 | size | What is the size of your clothing? | 2.1706 | 2 | 2.2038 | 2 | 1.6791 |
| 13 | cheating | How often have you cheated in exams? | 2.1185 | 2 | 2.3223 | 2 | 1.7319 |
| 14 | travel | Where would you like to travel? | 2.7488 | 2 | 1.5924 | 1 | 1.8464 |
| 15 | sex | How often do you have sex per week? | 1.8057 | 1 | 2.7867 | 3 | 1.9600 |
| 16 | weight | What is your weight? | 2.1043 | 2 | 2.7014 | 3 | 2.0284 |
| 17 | lying | How often do you lie to your best friend? | 2.3744 | 2 | 2.5308 | 2 | 2.0573 |
| 18 | drink | What is your favorite drink? | 2.8294 | 3 | 1.9668 | 2 | 2.0692 |
| 19 | actor | Who is your favorite actor? | 4.0616 | 4 | 2.3602 | 2 | 3.3502 |
| 20 | song | What is currently your favorite song? | 4.4692 | 5 | 2.1801 | 2 | 3.6644 |

"very comfortable" were coded with a value of 1, while "very hard" and "very uncomfortable" were coded with a value of 6. In Table 3, items are sorted on the basis of a combined measure of both evaluations, $\sqrt{(mean_{cognitive\ load} - 1)^2 + (mean_{emotional\ load} - 1)^2}$, calculated as the Euclidean distance from the most positive evaluation on both dimensions (i.e., from the origin (1, 1)).

## Procedure

We conducted two sessions for each of the four treatments in the *Cologne Laboratory for Economic Research (CLER)*, *University of Cologne*. We recruited our participants using the Online Recruitment System for Economic Experiments [14]. Overall, 212 subjects participated with 55% female and 45% male, and mostly participants were students from economics, social sciences, and business administration. Before the experiment, a random draw determined the order in which items were put into each subject's envelope in the *info* condition.

Each experimental session lasted about one hour. The written instructions in the *info* condition informed the subject that the experiment involves a decision to either keep information for her- or himself, or to truthfully answer the question and thereby provide it to the group account. In the *money* condition, the written instructions stated that the experiment involves a decision to either keep money for her- or himself or to provide it to the group account. In all treatments, subjects received 20 sheets of paper in an envelope. In the *info* condition, the term "information" and one of the 20 questions given in Table 3 were printed on each sheet of paper. To provide an item to the group account, the subject was asked to write the answer to the question on the respective sheet of paper. We emphasized that the answers should be truthful, but were kept strictly confident with regard to the other participants. Moreover, we ensured that answers to the questions could not be ascribed to a subject's identity. This was also stated clearly in the instructions. In the *money* condition, the term "money" and a text input field were printed on each sheet of paper. To provide a unit to the group account, the subject was asked to write the word "group" on the respective sheet of paper. The instructions stated clearly that no subject would receive feedback about individual provision levels. The instructions can be found in S2 File and the pictures of the experimental setup are provides in S1–S6 Figs.

After subjects made their provision decisions, they were asked to put all sheets back into the envelope and hand it over to the experimenter. While the experimenter calculated the payoffs in a separate room, subjects answered a short questionnaire (see S3 File). After the experiment, subjects were paid anonymously in a separate room. On average, participants earned € 10.20 in INFO (min: € 5.20; max: € 15.10), € 14.90 in INFO_NC (min: € 12.90; max: € 15.70), € 10.00 in MONEY (min: € 6.10; max: € 13.70), and € 15.50 in MONEY_NC (min: € 13.30; max: € 15.70) including the show-up fee of € 2.50. On average participants earned € 11.10 in the *cost* condition (minimum € 7.50, and maximum € 15.10), and € 17.40 in the *no cost* condition (minimum € 15.20, and maximum € 18.10) including a show up fee of € 2.50. The average earning was € 12.60 in the *info* condition (minimum € 5.10, and maximum € 15.70), and € 12.70 in the *money* condition (minimum € 6.10, and maximum € 15.70). We collected between 52 and 56 independent observations for each of the four treatments. By conducting one-shot experiments, we ensured statistical independence of all observations. If not stated otherwise, statistical comparisons between treatments are based on two-sided Mann-Whitney U tests (MWU), and comparisons within treatments are based on two-sided Wilcoxon signed-rank tests (WSR).

## Results

We present our experimental results in two steps. In section 5.1 we analyze treatment differences with regard to average provision levels. We find that (1) average contributions are significantly lower in presence of explicit net provision costs both in the *info* and the *money* condition, and (2) that the provision level in the *money* condition is higher than in the *info* condition if explicit net provision costs are zero. In section 5.2 we analyze the likelihood of item provision in the *info* condition. We find that (3) subjects engage in selective information provision, i.e., items with a low implicit provision cost are more likely provided than items with a high implicit provision cost. Finally, in section 5.3 we present the results of a follow-up experiment. This experiment is designed to test our finding (3) out of sample with new subjects. There we show that (4) when subjects are only confronted with items with low implicit provision costs, there is no difference between the provision of information and money, in particular there no longer is under-provision in case of zero explicit net costs.

### Treatment effects on information and money provision

In this section, we analyze the effect of explicit net provision costs on the provision of information and money. According to hypothesis 1, we expect higher average provision levels in the *no cost* condition than in the *cost* condition. According to hypothesis 2, we expect higher average provision levels in the *money* than in the *info* condition.

As shown in the first row of Table 4, subjects' average provision levels are close to 50% of the endowment, if provision is costly to them. On average, subjects contribute 9.23 items in INFO, and 8.12 units in MONEY. If contributions exhibit zero net costs, average levels are substantially higher. Subjects provide on average 17.80 items in INFO_NC and 19.29 units in MONEY_NC. With regard to the effect of explicit net provision costs on the provision levels, we find that average contributions are significantly lower in presence of explicit net provision costs both in the *info* and the *money* condition ($p = 0.0000$, for both comparisons). This finding supports hypothesis 1.

### Result 1

*With positive explicit net provision costs the provision level is lower than with zero explicit net provision costs, both in the info and the money treatments.*

With regard to differences between the provision of information and money, we find mixed results. Hypothesis 2 is rejected for the *cost* condition where we find no statistically significant difference between average provision levels (INFO vs. MONEY: $p = 0.5144$). However, we cannot reject hypothesis 2 for the *no cost* condition where subjects provide significantly more money than information (INFO_NC vs. MONEY_NC: $p = 0.0010$).

**Table 4. Average provision levels.**

|  | **INFO** | **INFO_NC** | **MONEY** | **MONEY_NC** |
|---|---|---|---|---|
| Average size of contributions (SD) | 9.23 (1.11) | 17.80 (0.61) | 8.12 (1.01) | 19.29 (0.47) |
| Ratio of contributions to endowment | 0.46 | 0.89 | 0.41 | 0.96 |
| Ind. obs. | 52 | 56 | 52 | 52 |

The table reports average provision levels and standard deviation (in parentheses) by treatment.

## Result 2

> *With positive explicit net provision costs, there is no difference in the provision of information and money. Yet, with zero explicit net provision costs, provision levels are higher in the money treatment than in the info treatment.*

To investigate the drivers of the second result, we examine the impact of the cognitive and emotional load on the likelihood of information provision in the next section.

## Selective information provision

In this section, we investigate the impact of implicit provision costs on the likelihood of information provision. There is a significant negative correlation between the item's combined measure of implicit provision costs and the respective average provision level in both treatments (Spearman's rank correlation coefficient, INFO: $p = 0.0013$; INFO_NC: $p = 0.0061$).

Table 5 reports the results from probit regressions (with the likelihood of item contribution as the dependent variable) to test how the implicit provision cost affects the likelihood of information provision. In model (1), the first coefficient replicates the above finding that explicit monetary net provision costs have a significant negative effect on the likelihood of item provision. The second coefficient shows that the higher an item scores on the combined cognitive and emotional dimension, the less likely it is to be provided. In model (2), we disentangle the effect of cognitive and emotional load. We find that while both coefficients are negative, only the mean emotional load has a significant negative effect on the likelihood of item provision (note that the two measures are significantly correlated (Spearman's rank correlation coefficient = 50.85%, $p = 0.0221$)). The significant negative effect of the emotional load and the combined measure yields support for hypothesis 3.

## Result 3

> *Subjects engage in selective information provision: The higher the item's implicit provision cost, the lower is the likelihood of the item's provision.*

**Table 5. Likelihood of item contribution.**

|  | (1) | (2) |
|---|---|---|
| explicit monetary provision cost | -1.3388*** | -1.3624*** |
| (0 = *no cost*, 1 = *cost*) | (0.22) | (0.22) |
| combined measure cognitive and emotional load | -0.1825*** |  |
| (Euclidean distance from origin) | (0.04) |  |
| mean cognitive load |  | -0.0239 |
| (1 = very low, 6 = very high) |  | (0.03) |
| mean emotional load |  | -0.6175*** |
| (1 = very low, 6 = very high) |  | (0.10) |
| constant | 1.5255*** | 2.5660*** |
|  | (0.18) | (0.29) |
| number of observations | 2,160 | 2,160 |
| number of subjects | 108 | 108 |
| Pseudo-R-squared | 0.1885 | 0.2010 |

Probit regression clustered by subject, robust standard errors in parentheses.

*: $p<0.1$

**: $p<0.05$

***: $p<0.01$. Dependent variable: Contribution of item (0 = no, 1 = yes).

**Table 6. Treatments of reduced question set experiment.**

|  | *info* | *money* |
|---|---|---|
| *cost* | INFO_RED | MONEY_RED |
| *no cost* | INFO_RED_NC | MONEY_RED_NC |

## Follow-up experiment: Information provision of items with low implicit costs

Our analysis uncovered the unexpected result that with zero explicit net provision costs, provision levels in the *info* condition are lower than in the *money* condition (cf. Result 2) and suggests that this under-provision is caused by implicit provision costs (cf. Result 3). To investigate the hypothesis that selective information provision caused by implicit costs may explain the lower level of information provision compared to money provision, we set up a follow-up experiment with a set of new subjects from the same subject pool as in our main experiment. In this new experiment, we replicated the design described above (see section 4.1), but with a *reduced* set of questions. We reduced the set of items in the *info* condition to the 10 questions with the lowest combined measure of cognitive and emotional load (see upper part of Table 3) and accordingly reduced the endowment in the *money* condition to 10 units. If it were true that implicit costs are the main driver for the under-provision of information, we should expect to see no significant difference between the money and the info treatment in the follow-up experiment.

The follow-up experiment comprises four treatments, summarized in Table 6.

To ensure payoff equivalence between all treatments the marginal payoffs are doubled (compare Table 7). All other parameters remain unchanged. The procedure in the reduced question set experiment followed the same protocol as in the four main treatments (see section 4.2). Overall, 212 subjects participated with 59% female and 41% male, and most participants were students from economics, social sciences and business administration. In the reduced question set experiment, on average participants earned €10.10 in the cost condition (minimum €5.10, and maximum €15.10), and €15.20 in the no cost condition (minimum €12.80, and maximum €15.70).

Overall, 212 subjects participated in the four treatments of the *reduced question set experiment*. To enable the statistical comparison between the four main treatments and the treatments of the *reduced question set experiment*, in the following we present provision levels as percentages of endowments (relative provision levels).

Table 8 contains three observations in support of our hypothesis that the high implicit provision costs induce the under-provision of items in INFO_NC as compared to the observed provision of money in MONEY_NC. First, we see that when the questions in the *info* condition only concern the items with the low implicit provision costs (INFO_RED_NC), relative provision levels are weakly significantly higher than in the full question set INFO_NC (p = 0.0998). Second, we observe that this is only true for information, but not for money: relative provision levels are not significantly different between MONEY_NC and MONEY_RED_NC (p = 0.4950). Third, when explicit net cost are zero, we do not observe the under-provision

**Table 7. Marginal payoffs.**

|  | *cost* condition | | *no cost* condition | |
|---|---|---|---|---|
|  | **self** | **others** | **self** | **others** |
| item/unit kept | €0.60 | €0.00 | €0.60 | €0.00 |
| item/unit provided | €0.24 | €0.24 | €0.60 | €0.24 |

**Table 8. Relative provision levels.**

| | info | | money | |
|---|---|---|---|---|
| **Ratio of contributions to endowment . . .** | *cost* | *no cost* | *cost* | *no cost* |
| . . . in the full set of 20 questions | 0.46 | 0.89 | 0.41 | 0.96 |
| . . . in the reduced set of 10 questions with low implicit costs | 0.46 | 0.93 | 0.33 | 0.98 |

(compared to money) as in the full question set: relative provision levels are not significantly different between INFO_RED_NC and MONEY_RED_NC (p = 0.2322).

## Result 4

*If subjects are only confronted with items with low implicit provision costs, there is no difference between the provision of information and money. In particular there no longer is underprovision in case of zero explicit net costs.*

In Table 9, we report the results from a probit regression that replicates the analysis of Table 5 in section 5.2. Again, the dependent variable is the likelihood of item provision. The results replicate our findings summarized in result 3 that subjects engage in selective information provision: The explicit monetary provision cost has a significant negative effect on the likelihood of item provision. Although subjects were only confronted with the 10 items with the lowest intrinsic costs, there is a variance in the intrinsic costs and the regression shows that the higher an item scores on the combined cognitive and emotional dimension, the less likely it is provided. Again, the emotional load has a significant negative effect on information provision, while the mean cognitive load is insignificant.

## Conclusion

In this paper, we experimentally study the provision of personal information to the benefit of others, and compare this to the provision of money. There is an important structural

**Table 9. Likelihood of item contribution–reduced question set experiment.**

| | (1) | (2) |
|---|---|---|
| explicit monetary provision cost | -1.5789*** | -1.5823*** |
| (0 = *no cost*, 1 = *cost*) | (0.26) | (0.26) |
| combined measure cognitive and emotional load | - 0.8531*** | |
| (Euclidean distance from origin) | (0.16) | |
| mean cognitive load | | 0.0414 |
| (1 = very low, 6 = very high) | | (0.12) |
| mean emotional load | | -1.1100*** |
| (1 = very low, 6 = very high) | | (0.22) |
| constant | 2.2751*** | 3.4461*** |
| | (0.30) | (0.49) |
| number of observations | 1,080 | 1,080 |
| number of subjects | 108 | 108 |
| Pseudo-R-squared | 0.2382 | 0.2400 |

Probit regression clustered by subject, robust standard errors in parentheses. *: p<0.1, **: p<0.05
***: p<0.01. Dependent variable: Contribution of item (0 = no, 1 = yes).

difference between money and information: Information provision exhibits an implicit cost that varies with the type of information. We account for this by including an additional cost parameter in our model and hypothesize that a players' decision to contribute to an information-based public good does not only depend on the explicit provision cost, but also on the implicit cost of information provision. This leads to different predictions for the provision of information-based public goods as compared to monetary public goods: If players' information sets strongly vary in implicit costs of information provision, information-based public goods will be underprovided as compared to material public goods. Moreover, selective information provision can be expected if the information requested is heterogeneous in implicit provision costs.

In a laboratory experiment, we test our hypotheses using a 2x2-design where we vary (1) the unit of provision (*info* vs. *money*), and (2) the explicit net provision costs (*cost* vs. *no cost*). We study real information provision, i.e., subjects provide real personal information about their own preferences, past behaviors or physical characteristics. In the two *info* treatments, we exogenously vary the cognitive and emotional load of the information we retrieve to induce different implicit costs of information provision. In line with the recent experimental literature (see [3–5, 9]), we observe selective information provision both in presence and absence of explicit net provision costs. This even leads to under-provision of information as compared to money provision when explicit net provision costs are zero. Furthermore, in line with the literature we observe that information provision varies with incentives, i.e., we observe more information provision if explicit net costs are absent (see [6–8]). An interesting future direction would be to examine how subjects hold each other responsible for under-provision of information if it is implicitly costly. [15] show that a majority of subjects do not hold others responsible for factors beyond individual control that hinder contributions to public goods. This could also be the case if implicit costs impede the provision of personal information. Then, one would observe lower punishment rates in an information-based public goods game than in a money-based public goods game.

While social interaction games with payoffs in other non-monetary dimensions have been studied (e.g. knowledge [16], time [17], pain [18–19]), to the best of our knowledge, we are the first to investigate information provision of personal information that creates public benefits. We show that even in an abstract laboratory setting where we guarantee for subjects' privacy and data protection, the implicit costs of information provision lead to selective contribution behavior. Compared to other non-monetary payoff dimensions that generally seem to induce higher levels of generosity than money [19], it seems clear that the level of provision of information public goods crucially depend on the cognitive and emotional costs that the contributors incur. Hence, there is an additional dimension complicating the design of mechanisms for the provision of information public goods in comparison to material public goods. These findings open avenues for future research on understanding and counteracting biases in the provision of information public goods.

## Supporting information

**S1 Fig. Envelope *info* treatments.**
(TIF)

**S2 Fig. Envelope *money* treatments.**
(TIF)

**S3 Fig. Information sheets (*main* treatments).**
(TIF)

**S4 Fig. Information sheets (*reduced* treatments).**
(TIF)

**S5 Fig. Money sheets (*main* treatments).**
(TIF)

**S6 Fig. Money sheets (*reduced* treatments).**
(TIF)

**S1 File. Questionnaire of the survey study (translated from German).**
(DOCX)

**S2 File. Instructions (translated from German).**
(DOCX)

**S3 File. Post-experimental questionnaire (translated from German).**
(DOCX)

## Author Contributions

**Conceptualization:** Bettina Rockenbach, Abdolkarim Sadrieh, Anne Schielke.

**Data curation:** Anne Schielke.

**Formal analysis:** Bettina Rockenbach, Abdolkarim Sadrieh, Anne Schielke.

**Funding acquisition:** Bettina Rockenbach.

**Investigation:** Bettina Rockenbach, Abdolkarim Sadrieh, Anne Schielke.

**Methodology:** Bettina Rockenbach, Abdolkarim Sadrieh, Anne Schielke.

**Project administration:** Anne Schielke.

**Resources:** Bettina Rockenbach.

**Supervision:** Bettina Rockenbach, Abdolkarim Sadrieh.

**Validation:** Bettina Rockenbach, Abdolkarim Sadrieh, Anne Schielke.

**Writing – original draft:** Bettina Rockenbach, Abdolkarim Sadrieh, Anne Schielke.

**Writing – review & editing:** Bettina Rockenbach, Abdolkarim Sadrieh, Anne Schielke.

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
