## [Decision Letter · Decision Letter 0]

21 May 2020

PONE-D-20-09566

Providing Personal Information to the Benefit of Others

PLOS ONE

Dear Dr Rockenbach,

Thank you for submitting your manuscript to PLOS ONE. After careful consideration, we feel that it has merit but does not fully meet PLOS ONE’s publication criteria as it currently stands. Therefore, we invite you to submit a revised version of the manuscript that addresses the points raised during the review process. As you will see the referees are fairly different. While Reviewer #1 is very positive, Reviewer #2 is seriously concerned about the external validity of your experiment. I found myself the paper also interesting and I decided to give it a chance. I think that a nice revision might alleviate referee #2. However you should be aware that I will send the paper back to both referees.

We would appreciate receiving your revised manuscript by Jul 05 2020 11:59PM. To enhance the reproducibility of your results, we recommend that if applicable you deposit your laboratory protocols in protocols.io, where a protocol can be assigned its own identifier (DOI) such that it can be cited independently in the future. For instructions see: http://journals.plos.org/plosone/s/submission-guidelines#loc-laboratory-protocols

We look forward to receiving your revised manuscript.

Kind regards,

Pablo Brañas-Garza, PhD Economics

Academic Editor

PLOS ONE

Journal Requirements:

2. Please amend the title page within your main document to clearly indicate the corresponding author.

3. Please ensure that you refer to Figures A1-A6 in your text as, if accepted, production will need this reference to link the reader to each figure.

Reviewers' comments:

Reviewer's Responses to Questions

**Comments to the Author**

1. Is the manuscript technically sound, and do the data support the conclusions?

Reviewer #1: Yes

Reviewer #2: Partly

2. Has the statistical analysis been performed appropriately and rigorously? 

Reviewer #1: Yes

Reviewer #2: Yes

3. Have the authors made all data underlying the findings in their manuscript fully available?

Reviewer #1: Yes

Reviewer #2: Yes

4. Is the manuscript presented in an intelligible fashion and written in standard English?

Reviewer #1: Yes

Reviewer #2: Yes

5. Review Comments to the Author

Reviewer #1: Providing personal information to the benefit of others

Referee Report

*** Summary ***

The authors report the results of two public goods games. The first one is the standard version of the game, in which four players must decide how many of their 20 units of endowment they want to contribute to a common fund, granting higher earnings to the group, and how many they want to keep for themselves. The novelty in this design is a treatment in which participants do not contribute monetary tokens, but rather write down and contribute personal information details (20 different questions), varying in cognitive costs (associated to recall), and in emotional costs. In the MONEY and INFO treatments, there is also an exogenous variation in the cost of contribution (POSITIVE vs ZERO cost). The authors find lower provision in the INFO game compared to the standard MONEY game. They also find that implicit costs (cognitive and emotional) of specific information are correlated with the willingness to provide that specific information.

*** General comments ***

What I like the most about the paper is the contribution on bringing the “cooperation to a public good” component to the important debate on the disclosure and handling of personal information. The authors are aware of the relevant literature, and it helps a lot the reader to understand the aimed contribution of this study. The authors also do a good job in highlighting the advantages of bringing this problem to the lab (provided information is secured, and there should be lower trust concerns on how information will be handled).

My main concern is the connection between the very interesting examples provided in the abstract and in the introduction (traffic flows and health issues), and the type of requested information in the experiment. I understand that, as an experimentalist, sometimes one needs to sacrifice some external validity to gain control and internal validity. However, in the proposed design, I am afraid that the type of requested information, and the process to collect it, does not necessarily evoke the nature of “public good” that the disclosed information might have.

This might be problematic for two reasons. First, if the reason why my private information contributes to a greater knowledge is not clear, I have less incentives to truthfully report. In other words, I can contribute to the group, even providing false information (which mechanically can also happen in your game). Second, in the same way that you model an additional cost Gamma capturing implicit individual costs, one could think of an additional parameter (let’s call it Phi) that might capture other non-monetary costs (or benefits) of the aggregated information. The disclosed or “shared” information might trigger other type of behaviors in which I compare my information with the rest of the group (e.g., conformity).

This concern does not invalidate the experiment. But I would like to see a discussion on this issue.

*** Other concerns/questions/comments ***

1. What do “social preferences” mean in this context?

It is not clear if in the context of information disclosure, social preferences include more than the usual altruism and reciprocity concerns. For instance, shame or (dis-)conformism.

In the same line, it is not clear what would be here to “maximize joint utility.”

2. On the decision of making provision cost zero

I do not think this is necessarily a bad decision, but as reader I would like to understand better why implement a cost zero, rather than a very small cost (e.g., €0.01).

3. On the recruitment

Participants of this experiment had previous experience in this Lab? Experienced participants might be more aware of protocols for handling information, and therefore they might face lower psychological costs of providing information. I know that this feature will be balanced across treatments, but it might be useful as a control variable in case you have a combination of new and experienced lab participants.

4. On the confounding between cognitive and emotional costs

A different way to conduct the experiment would have been to ask all the participants to write their responses to the 20 questions, and then “destroy/tear/hide” those papers with information they were not willing to reveal. The main advantage would have been that cognitive costs associated to information recalling would have not played a role, and it would have been possible to focus on emotional costs. Moreover, I think (but you may disagree) that it would also allow to reduce misreport of information.

I would like to know whether you considered a design with these features, but you might have discarded for some other reasons that I am not seeing (e.g., ethical or methodological issues).

*** Minor comments ***

• The literature review is too detailed, you could shorten this section.

• Lines 314-316: Please provide a similar line for the comparison INFO vs. MONEY

• Lines 323-324: This sentence is hard to follow. Please rewrite.

Reviewer #2: Review report for manuscript “Providing Personal Information to the Benefit of Others “

This paper reports on a laboratory experiment that investigates contribution behavior based on personal information. The experimental design also involves treatments based on standard monetary incentives which makes comparison of two cases possible. The results indicate that contribution decisions in personal information treatments are related to emotional and cognitive costs; suggesting a conflict between standard public good behavior and information-based public goods.

This has not been one of the easiest reviews I have done. On the one hand, the research question is very relevant especially nowadays (with Covid-19 and also digitalization). The manuscript is decently-written and the experiment was well conducted. On the other hand, I found the experimental design too-disconnected from the real world and therefore artificial. Laboratory is ideal for investigating some specific research questions, but for a research question like this I do not think it is sufficient. It is true that the experiment is interesting, but a standard lab experiment does not seem enough to be enough to provide a strong evidence. Today, there are countless options to ask public opinion on a topic like this (online platforms, household surveys etc.)- and I am surprised the authors did not try to extend their study. In a nutshell, the study suffers from external validity problem strongly in my humble opinion.

Some specific comments:

1- The personal information shared by the participants/survey respondents are not even similar to those individuals consent in real life. Most importantly, individuals’ aversion to share personal information in the real world is strongly connected to reliability of institutions and decision makers’ trust levels in institutions. The conceptual framework simplifies the mechanism too much. Furthermore, when individuals give consent on using their personal information (such as internet browsing or health history), they cannot lie about their personal data. Because most of the time, algorithms and technologies automatically retrieve the information. Or if it is a public institution that collects the data, individuals refrain from lying. In the experiment, the instructions are asking the respondents to be honest, which does not seem to be enough.

2- The paper can benefit from the literature on private-collective innovation. For example:

Gächter, S., von Krogh, G., & Haefliger, S. (2010). Initiating private-collective innovation: The fragility of knowledge sharing. Research Policy, 39(7), 893-906.

and studies of Eric von Hippel

Also the literature on experiments comparing decisions with monetary and non-monetary goods/bads. Some examples:

Davis, A. L., Miller, J. H., & Bhatia, S. (2018). Are preferences for allocating harm rational?. Decision, 5(4), 287.

Erkut, H. (2018). Social norms and preferences for generosity are domain dependent. WZB Discussion Paper.

Noussair, C.N. and Stoop, J., 2015. Time as a medium of reward in three social preference experiments. Experimental Economics, 18(3), pp.442-456.

Story, G. W., Vlaev, I., Metcalfe, R. D., Crockett, M. J., Kurth-Nelson, Z., Darzi, A., & Dolan, R. J. (2015). Social redistribution of pain and money. Scientific reports, 5, 15389.

3- Implicit costs are not explained well. For example, cognitive cost is vaguely mentioned in page 3. But what does this correspond to in the real-world decision making? Why is it an important piece of the experimental design?

4- While the literature review section reviews the literature in detail (maybe even too detailed), the introduction cannot place the current study in the current literature and motivate the paper well. In other words, literature review and motivation should be better connected. As it is the literature review seems like a very separate section and the manuscript does not help reader to understand where this study aims to stand in the literature.

5- Considering the facts that there are 4 members in each group, the total sample size is not too large and the experiment was not pre-registered, the reader gets curious about the power analysis.

All in all, although I find the topic very interesting, I do not find the manuscript convincing in its current form.

Kind Regards

6. PLOS authors have the option to publish the peer review history of their article (what does this mean?). If published, this will include your full peer review and any attached files.

Reviewer #1: No

Reviewer #2: No

---

## [Author Response · Author response to Decision Letter 0]

3 Jul 2020

The detailed responses to the reviewers are included in the file "Response to Reviewers".

---

## [Decision Letter · Decision Letter 1]

22 Jul 2020

Providing personal information to the benefit of others

PONE-D-20-09566R1

Dear Dr. Rockenbach,

We’re pleased to inform you that your manuscript has been judged scientifically suitable for publication and will be formally accepted for publication once it meets all outstanding technical requirements.

Kind regards,

Pablo Brañas-Garza, PhD Economics

Academic Editor

PLOS ONE

Additional Editor Comments (optional):

Reviewers' comments:

Reviewer's Responses to Questions

**Comments to the Author**

1. If the authors have adequately addressed your comments raised in a previous round of review and you feel that this manuscript is now acceptable for publication, you may indicate that here to bypass the “Comments to the Author” section, enter your conflict of interest statement in the “Confidential to Editor” section, and submit your "Accept" recommendation.

Reviewer #1: All comments have been addressed

Reviewer #2: All comments have been addressed

2. Is the manuscript technically sound, and do the data support the conclusions?

Reviewer #1: Yes

Reviewer #2: Yes

3. Has the statistical analysis been performed appropriately and rigorously? 

Reviewer #1: Yes

Reviewer #2: Yes

4. Have the authors made all data underlying the findings in their manuscript fully available?

Reviewer #1: Yes

Reviewer #2: Yes

5. Is the manuscript presented in an intelligible fashion and written in standard English?

Reviewer #1: Yes

Reviewer #2: Yes

6. Review Comments to the Author

Reviewer #1: Thank you for responding to all my comments. I am satisfied with the provided explanations, specially those on why not collecting all the personal information and make the decision about which information to share.

Reviewer #2: Dear authors,

I am well aware that my previous report involves mostly structural criticisms and not possible to address without more data collection. Therefore, I will loosen my requests as I see that you did your best with the current data available.

I only request an additional extended discussion regarding the drawbacks of your study. Then the paper can be ready to be published. I do not require another round of revision.

Kind Regards

7. PLOS authors have the option to publish the peer review history of their article (what does this mean?). If published, this will include your full peer review and any attached files.

Reviewer #1: No

Reviewer #2: No

---

## [Editor Report · Acceptance letter]

27 Jul 2020

PONE-D-20-09566R1 

Providing personal information to the benefit of others 

Dear Dr. Rockenbach:

I'm pleased to inform you that your manuscript has been deemed suitable for publication in PLOS ONE. Congratulations! Your manuscript is now with our production department. 

Kind regards, 

on behalf of

Dr Pablo Brañas-Garza 

Academic Editor

PLOS ONE